# Arena: A Scalable and Configurable Benchmark for Policy Learning

**Shuang Liu\*[1], Sirui Xu\*[2], Tongzhou Mu[1], Zhiwei Jia[1], Yiran Wu[1], Hao Su[1]**
[1] Department of Computer Science and Engineering, UC San Diego
[2] School of Electrical Engineering and Computer Science, Peking University

## Abstract

We believe current benchmarks for policy learning lack two important properties: scalability and configurability. The growing literature on modeling policies as graph neural networks calls for an object-based benchmark where the number of objects can be arbitrarily scaled and the mechanics can be freely configured. We introduce the Arena benchmark[1], a *scalable* and *configurable* benchmark for policy learning. Arena provides an object-based game-like environment where the number of objects can be arbitrarily *scaled* and the mechanics can be *configured* with a large degree of freedom. In this way, arena is designed to be an all-in-one environment that uses scaling and configuration to smoothly interpolates multiple dimensions of decision making that require different degrees of inductive bias.

## 1 Introduction

Policy learning refers to the process of using machine learning techniques such as reinforcement learning (RL) and imitation learning (IL) to obtain a policy for sequential decision making. The past decade has witnessed a rapid growth of benchmarks for policy learning (Bellemare et al., 2013; Duan et al., 2016; Brockman et al., 2016; Beattie et al., 2016; Vinyals et al., 2017; Juliani et al., 2018; 2019; Yu et al., 2019; Guss et al., 2019; Cobbe et al., 2020; Tassa et al., 2020; Toyer et al., 2020). However, these benchmarks lack two important properties: *scalability* and *configurability*. For example, Atari 2600 games (Bellemare et al., 2013; Brockman et al., 2016; Machado et al., 2018) are among the most used benchmarks for RL and IL (Jaderberg et al., 2016; Horgan et al., 2018; Kapturowski et al., 2018; Hessel et al., 2018; Espeholt et al., 2018; Schmitt et al., 2020; Schrittwieser et al., 2020); however, these games are essentially black boxes where there is no way to change the size of the map, the number of objects, or the game dynamics. This is problematic, since on these games *learning* may degenerate to *memorizing* the specific positions and properties of the objects. More recent benchmarks address this problem by model a game as an instance drawn from a population of similar games and perform training on this population (Justesen et al., 2018; Cobbe et al., 2020; Toyer et al., 2020). Although the training population does introduce some variances in map sizes, quantities of objects, and game dynamics, the distribution of these variances are dictated by the benchmarks and are constrained within a small range.

Why do we need a scalable and configurable benchmark for policy learning? In the past few years, there has been a growing interest in learning *scalable functions* (Gilmer et al., 2017; Selsam et al., 2019; Tang et al., 2020; Yehudai et al., 2021) and *scalable policies* (Dai et al., 2017; Yolcu and Póczos, 2019; Mu et al., 2020; Tang et al., 2020) using graph neural networks (GNNs) (Scarselli et al., 2008). However, the benchmarks used in these papers are either not particularly suitable for policy learning (e.g. pure combinatorial problems), or not fully scalable. Furthermore, many of the current policy

---

[1] Benchmark website: https://github.com/Sirui-Xu/Arena

Submitted to the 35th Conference on Neural Information Processing Systems (NeurIPS 2021) Track on Datasets and Benchmarks. Do not distribute.

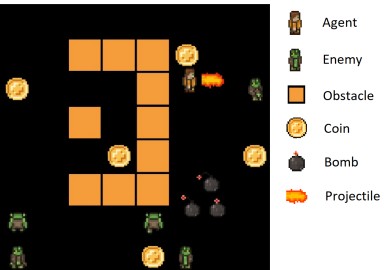

(a) A state in a small-scale Arena environment rendered as an image.

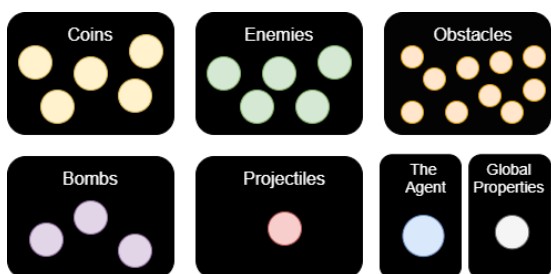

(b) A graphical illustration of the **state representation of (a)** for policy learning (for details see Section 3.1).

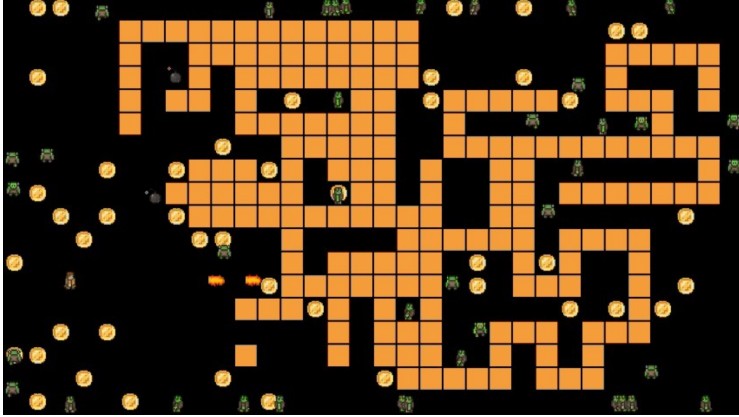

(c) A state in a large-scale Arena environment rendered as an image.

Figure 1: Visualizing states of the Arena environments. An instance of the Arena benchmark starts with an arbitrarily sized region (i.e., the arena) containing a controllable *agent* as well as an arbitrary number of destructable *obstacles*, *enemies*, and collectable *coins*. The agent can move in four directions, fire *projectiles*, as well as place *bombs*. The goal is to control the agent to collect as many coins as possible in the shortest amount of time, potentially kill enemies and destroy obstacles using the projectiles and bombs along the way.

learning methods are deeply coupled with machine perception to utilize the well-known prowess of convolutions neural networks (CNNs). Such coupling could be problematic since it becomes very difficult to tell whether a good policy learning algorithm is better at policy learning or simply at object detection. Since CNNs only have fixed receptive field and depth, a scalable benchmark may potentially force the policy learning to be decoupled from machine perception by introducing long-range relations. Configurablity is also crucial, for policy learning is a broad concept that can be as low-level as robot-arm manipulation, or as high-level as causal inference. Ideally, a benchmark or a suite of benchmarks should have the granularity to selectively target different levels of decision making.

We introduce Arena, a scalable and configurable benchmark for policy learning (Figure 1). It is an object-based game-like environment. The game logic is reminiscent of many classic games such as Pac-Man and Bomberman. An instance of the Arena benchmark starts with an arbitrarily sized region (i.e., the arena) containing a controllable agent as well as an arbitrary number of destructable obstacles, enemies, and collectable coins. The agent can move in four directions, fire projectiles, as well as place bombs. The goal is to control the agent to collect as many coins as possible in the shortest amount of time, potentially kill enemies and destroy obstacles using the projectiles and bombs along the way.

The Arena benchmark achieves scalability by using object-based state representations. For example, the obstacles in the arena are represented by a set of tuples $(x, y, s)$ where $(x, y)$ is the coordinate of an obstacle and $s$ is its size. Similar representations are used for enemies, coins, bombs, and projectiles. Object-based state representations can contain an arbitrary number of objects, thus an environment can be "scaled up" by adding more objects to it.

The Arena benchmark achieves configurability by allowing a much larger degree of freedom in altering different aspects of the environment compared to existing benchmarks. For example, the default procedure for determining the initial coordinates of the agent, the enemies, the obstacles, and the coins can be modified through parameters, or be entirely overridden by the user; the default movement logic of enemies can be controlled by parameters or entirely overridden by the user; the number of obstacles can be set to 0 to put less emphasis on the routing capacity of the learned policy; the movement speed of the enemies can be set to 0 to reduce the difficulty so that the learned policy only have to know how to *avoid* instead of how to *evade*.

We will give a brief overview of the existing benchmarks for policy learning in Section 2, formally describe the Arena benchmark in Section 3, examine the difficulty of the benchmark in different configurations in Section 4, and discuss some preliminary experiments in Section 5.

## 2   Related Work

The most widely-used benchmarks for policy learning are perhaps the the Arcade Learning Environment (ALE) (Bellemare et al., 2013; Brockman et al., 2016; Machado et al., 2018) and MuJoCo environments (Duan et al., 2016; Brockman et al., 2016). ALE is an object-oriented framework that allows researchers and hobbyists to develop AI agents for Atari 2600 games. It is built on top of the Atari 2600 emulator Stella (Mott et al., 1995) and separates the details of emulation from agent design. Our Arena can be seen as a scalable and configurable version of ALE, although the sheer amount of games included in ALE covers a broader spectrum of game logic. MuJoCo environments refer to various tasks built upon the MuJoCo physics engine (Todorov et al., 2012), which is itself tailored to model-based control.

Other widely used benchmarks include: DeepMind Lab (Beattie et al., 2016), which is a 3D learning environment based on id Software's Quake III Arena via ioquake3 and other open source software. The StarCraft II Learning Environment (Vinyals et al., 2017), which is built upon a StarCraft II API that provides full external control of the video game StarCraft II. The Unity learning platform (Juliani et al., 2018), which is built upon the Unity engine. The Obstacle Tower (Juliani et al., 2019), which is a procedurally generated environment consisting of multiple floors to be solved by a learning agent. Meta-World (Yu et al., 2019), which is a simulated benchmark for meta-reinforcement learning and multi-task learning consisting of distinct robotic manipulation tasks. MineRL (Guss et al., 2019), which is built upon the video game Minecraft. The Procgen Benchmark (Cobbe et al., 2020), which is a suite of procedurally-generated environments which provide a direct measure of how quickly a reinforcement learning agent learns generalizable skills. DM_Control (Tassa et al., 2020), which is a software stack for physics-based simulation and reinforcement learning environments using MuJoCo physics. The Magical benchmark (Toyer et al., 2020), which is a benchmark suite for robust imitation learning.

## 3   The Arena Environment

The Arena environment has a python interface similar to the one provided in OpenAI Gym (Brockman et al., 2016). However, unlike in usual Gym-like environments where the observation/state space is either image-based or vector-based, the state space in Arena is object-based. The state transitions are markovian and the goal is to maximize the cumulated score from collecting coins, whose values decay at a certain rate over time.

### 3.1   State Representation

Each state in Arena is represented by a JSON-like structured data with no fixed size. Specifically, each state is a pair (GLOBAL, LOCAL) where GLOBAL is a python dictionary containing variables that does not change after a state transition, and LOCAL is a python dictionary containing variables that may change after a state transition. The details of the two dictionaries are described in Table 1 and Table 2, respectively.

Table 1: Specification of the GLOBAL dictionary in the state representation.

| Variable | Type | Range | Meaning |
|---|---|---|---|
| $H$ | float | $(0, \infty)$ | height of the map |
| $W$ | float | $(0, \infty)$ | width of the map |
| $S$ | float | $(0, \infty)$ | size of non-obstacle objects |
| $N_{\text{bomb}}$ | int | $\mathbb{N}$ | maximum number of bombs that can exist in the map |
| $N_{\text{projectile}}$ | int | $\mathbb{N}$ | maximum number of projectiles that can exist in the map |
| $D_{\text{bomb}}$ | int | $\mathbb{N}$ | number of steps a bomb remains in the map before it explodes |
| $r$ | float | $(0, \infty)$ | explosion radius of the bombs |
| $v_{\text{agent}}$ | float | $(0, \infty)$ | travel speed of the agent |
| $v_{\text{projectile}}$ | float | $(0, \infty)$ | travel speed of the projectiles |
| $v_{\text{enemy}}$ | float | $[0, \infty)$ | travel speed of the enemies |
| $\delta$ | float | $[0, 1]$ | probability each enemy changes direction in each step |
| $\gamma$ | float | $[0, 1)$ | rate at which the value of each coin decays |

Table 2: Specification of the LOCAL dictionary in the state representation.

| Variable | Type | Range | Meaning |
|---|---|---|---|
| $\mathcal{A}$ | tuple | $(x \in \mathbb{R}, y \in \mathbb{R}, d \in \{\text{L}, \text{R}, \text{U}, \text{D}\})$ | $(x, y)$ is the coordinate of the agent and $d$ is the direction it is facing |
| $\mathcal{B}$ | multiset | $\{(x \in \mathbb{R}, y \in \mathbb{R}, n \in \mathbb{N}\}$ | $(x, y)$ is the coordinate of a bomb and $n$ is the number of steps before it explodes |
| $\mathcal{P}$ | multiset | $\{(x \in \mathbb{R}, y \in \mathbb{R}, d \in \{\text{L}, \text{R}, \text{U}, \text{D}\}\}$ | $(x, y)$ is the coordinate of a projectile and $d$ is the direction it is facing |
| $\mathcal{E}$ | multiset | $\{(x \in \mathbb{R}, y \in \mathbb{R}, d \in \{\text{L}, \text{R}, \text{U}, \text{D}\}\}$ | $(x, y)$ is the coordinate of an enemy and $d$ is the direction it is facing |
| $\mathcal{C}$ | multiset | $\{(x \in \mathbb{R}, y \in \mathbb{R}), v \in [0, \infty)\}$ | $(x, y)$ is the coordinate of a coin and $v$ is its value |
| $\mathcal{O}$ | set | $\{(x \in \mathbb{R}, y \in \mathbb{R}, s \in (0, \infty)\}$ | $(x, y)$ is the coordinate of an obstacle and $s$ is its size |

## 3.2 Dynamics

The agent, the bombs, the projectiles, the enemies, the obstacles, the rewards are all counted as objects. Each object with coordinate $(x, y)$ and a size $s$ occupies the area $(x-s/2, x+s/2) \times (y-s/2, y+s/2)$. We say an object is *inside the map* if the region it occupies is contained in $[0, W] \times [0, H]$, and *outside the map* otherwise. We say one object *collides* with another object if their occupied areas have a non-empty intersection.

In each state, the player can take an action $a \in \{\text{L}, \text{R}, \text{U}, \text{D}, \text{SHOOT}, \text{BOMB}, \text{NONE}\}$ and transitions to either a *regular* state (as described in Section 3.1) or a *terminal* state, in which case the game ends. We now describe how the next state is derived from the current state if it is not a terminal state, and in which cases it is a terminal state. The following descriptions are meant to be interpreted *procedurally* — they behave like a program, later descriptions are based on previous descriptions.

**Notations.** For any $d \in \{\text{L}, \text{R}, \text{U}, \text{D}\}$, and type $\in \{\text{agent, projectile, enemy}\}$ if $d = \text{L}$, let $\Delta_{x,d,\text{type}} = -v_{\text{type}}$, $\Delta_{y,d,\text{type}} = 0$; if $d = \text{R}$, let $\Delta_{x,d,\text{type}} = v_{\text{type}}$, $\Delta_{y,d,\text{type}} = 0$; if $d = \text{U}$, let $\Delta_{x,d,\text{type}} = 0$, $\Delta_{y,d,\text{type}} = v_{\text{type}}$; if $d = \text{D}$, let $\Delta_{x,d,\text{type}} = 0$, $\Delta_{y,d,\text{type}} = -v_{\text{type}}$.

**The agent.** Suppose currently $\mathcal{A} = (x, y, d)$. If $a \in \{\text{L}, \text{R}, \text{U}, \text{D}\}$, then $\mathcal{A}$ becomes $(x + \Delta_{x,a,\text{agent}}, y + \Delta_{y,a,\text{agent}}, a)$. If at the new coordinate the agent would collide with any obstacle or be outside the map, the action is invalidated and overridden to NONE. If at the new coordinate the agent would collide with any coin $(x, y, v)$ in $\mathcal{C}$, remove the coin from $\mathcal{C}$ and a score of $v$ is accumulated. If $a = \text{SHOOT}$ and $|\mathcal{P}| < N_{\text{projectile}}$, then $(x + \Delta_{x,a,\text{projectile}}, y + \Delta_{y,a,\text{projectile}}, a)$ is added to $\mathcal{P}$. If $a = \text{BOMB}$ and $|\mathcal{B}| < N_{\text{bomb}}$, then $(x, y, D_{\text{bomb}})$ is added to $\mathcal{B}$.

**The enemies.** The default behavior of the enemies can be overridden by the user. The default behavior is as follows: for each enemy $(x, y, d)$ in $\mathcal{E}$, for any $d' \in \{\text{L}, \text{R}, \text{U}, \text{D}\}$, we say $d'$ is plausible if $(x + \Delta_{x,d',\text{enemy}}, y + \Delta_{y,d',\text{enemy}})$ would not collide any obstacle or be outside the map. Let $D$ be the set containing all the plausible $d'$. If $D$ is not empty, let $d'$ be drawn uniformly at random

from $D$ and $\iota$ be drawn uniformly at random from $[0,1]$, if $\iota < \delta$ or $d \notin D$, $(x,y,d)$ becomes $(x + \Delta_{x,d',\text{enemy}}, y + \Delta_{y,d',\text{enemy}}, d')$, otherwise $(x,y,d)$ becomes $(x + \Delta_{x,d,\text{enemy}}, y + \Delta_{y,d,\text{enemy}}, d)$. If the new coordinate of any enemy would collide with the new coordinate of the agent, the next state is the terminal state.

**The bombs.** For each bomb $(x,y,n) \in \mathcal{B}$, if $n < D_{\text{bomb}}$, $n$ becomes $n + 1$, otherwise, the bomb is removed from $\mathcal{B}$ and any object except coins with coordinate $(x',y')$ such that $|x - x'| + |y - y'| \leq r$ is *removed*. If the agent is removed, the next state is the terminal state; if any other object is removed, it is removed from the (multi)set it was in.

**The projectiles.** For each projectile $(x,y,d) \in \mathcal{P}$, it becomes $(x + \Delta_{x,d,\text{projectile}}, y + \Delta_{y,d,\text{projectile}}, d)$. If at the new coordinate the projectile is outside the map, the projectile is removed from $\mathcal{P}$; otherwise: if the at the new coordinate the projectile would collide with any object except coins, that object is *removed*. If the agent is removed, the next state is the terminal state; if any other object is removed, it is removed from the (multi)set it was in.

**The coins.** After each step, for each coin $(x,y,v)$ that was not removed from $\mathcal{C}$, its $v$ becomes $\gamma \cdot v$.

### 3.3 Default Object Generation

We will describe how the objects in the initial state are generated by default given the number of obstacles $N_{\text{obstacle}}$, obstacle size $S_{\text{obstacle}}$, the number of enemies $N_{\text{enemy}}$, and the number of coins $N_{\text{coin}}$. Of course, this procedure can be overriden by the user, as long as the following requirements are satisfied: In the initial state: $|\mathcal{B}| = |\mathcal{P}| = 0$; all objects are inside the map; the agent shall not collide with any other object except bombs; no obstacle shall collide with any other object. Starting with a blank arena $[0,W] \times [0,H]$, all the objects are generated sequentially as described below:

**Generating obstacles $\mathcal{O}$.** If $N_{\text{obstacle}} \geq 1$, then the first obstacle is sampled uniformly at random so that it is inside the arena. Then the following procedure is repeated until a total of $N_{\text{obstacle}}$ obstacles have been generated: Suppose the last generated obstacle's coordinate is $(x,y)$, let $D = \{(x - 2S_{\text{obstacle}}, y), (x + 2S_{\text{obstacle}}, y), (x, y - 2S_{\text{obstacle}}), (x, y + 2S_{\text{obstacle}})\}$. Let $D'$ contain all elements $(x,y)$ of $D$ such that an obstacle at placed at $(x,y)$ is not outside the arena. Choose $(x',y')$ randomly from $D'$. If no obstacle has been generated at $(x',y')$, generated one at $(x',y')$ with size $S_{\text{obstacle}}$. If no obstacle has been generated at $((x+x')/2, (y+y')/2)$, generated one at $((x+x')/2, (y+y')/2)$ with size $S_{\text{obstacle}}$.

**Generating enemies $\mathcal{E}$.** The following procedure is repeated $N_{\text{enemy}}$ times: an enemy is generated at a coordinate uniformly at random so that it is inside the arena and does not collide with any obstacles, its direction is chosen uniformly at random.

**Generating coins $\mathcal{C}$.** The following procedure is repeated $N_{\text{coin}}$ times: a coin is generated at a coordinate uniformly at random so that it is inside the arena and does not collide with any obstacles, its value is 1.

**Generating the agent $\mathcal{A}$.** The coordinate of the agent is chosen uniformly at random so that the agent is inside the arena and it does not collide with any obstacle, enemy, or coin. The direction of the agent is chosen uniformly at random.

### 3.4 Simulation and Rendering

The speed of simulation and rendering largely depends on the number of object in the arena. We provide some crude statistics: On a hexa-core Intel(R) Core(TM) i7-6850K CPU with 3.60GHz, the number of steps per second is $1.8 \times 10^4$ when simulating 10 objects, and $2.3 \times 10^3$ when simulating 100 objects; the number of steps per second is $1.2 \times 10^4$ when simulating 10 objects and redering with the DRAW module in PyGame, and $2.0 \times 10^3$ when simulating and rendering 100 objects.

## 4 Configuring Arena

Arena can be configured to test different dimensions of decision making. Configuration refers to the process of specifying the generation process of the initial state (i.e. everything in Table 1 and Table 2) as well as the movement logic of the enemies. Therefore, each configuration corresponds to a distribution of Markov Decision Processes (MDPs).

In this section, we will discuss the following configurable elements: $N_{\text{bomb}}$, $N_{\text{projectile}}$, $v_{\text{enemy}}$, $N_{\text{obstacle}}$ — the size of $\mathcal{O}$ in the initial state, $N_{\text{enemy}}$ — the size of $\mathcal{E}$ in the initial state, $N_{\text{coin}}$ — the size of $\mathcal{C}$ in the initial state, and the movement protocol of the enemies. We believe these elements have significant impact on the level of inductive bias required to learn a good policy.

We introduce some representative configurations, listed below. The naming convention is as follows: Configurations without **X** in the name has only one coin, those with **X** in the name has more than one coin. As the the number of coins increases, the problem starts to involve a variant of the traveling salesman problem (TSP) and theoretically calculating the optimal policy becomes intractable. Configurations starting with **A** does not have moving enemies and does not involve bombs and projectiles; configurations starting with **B** adds moving enemies but still does not involve bombs and projectiles; configurations starting with **C** have moving enemies and also involve bombs and projectiles. Configurations ending with **0** does not have obstacles or enemies; configurations ending with **1** add obstacles but still do not have enemies; configurations ending with **2** have both obstacles and non-movable enemies.

**A0.** Let us start from the simplest case: $N_{\text{enemy}} = 0$, $N_{\text{obstacle}} = 0$, $N_{\text{coin}} = 1$. In this case, the optimal policy is a greedy policy that simply moves the agent towards the only coin following the Manhattan distance between them. The policy can be calculated in time $O(1)$.

**A1.** A slightly more complicated case is $N_{\text{enemy}} = 0$, $N_{\text{obstacle}} > 0$, $N_{\text{coin}} = 1$, $N_{\text{bomb}} = 0$, $N_{\text{projectile}} = 0$. A good policy for this scenario needs to have basic routing capacities. In fact, the optimal policy corresponds to a shortest path problem where the underlying graph has size $O(N_{\text{obstacle}})$. Therefore, the optimal policy requires time $O(N_{\text{obstacle}} \cdot \log(N_{\text{obstacle}}))$ to calculate.

**A2.** A even more complicated, but still manageable case is $N_{\text{enemy}} > 0$, $v_{\text{enemy}} = 0$, $N_{\text{obstacle}} > 0$, $N_{\text{coin}} = 1$, $N_{\text{bomb}} = 0$, $N_{\text{projectile}} = 0$. A good policy for this scenario needs to have basic routing capacities, and also need to be aware of avoiding enemies. In fact, the optimal policy corresponds to a shortest path problem where the underlying graph has size $O(N_{\text{obstacle}} + N_{\text{enemy}})$. Therefore, the optimal policy requires time $O((N_{\text{obstacle}} + N_{\text{enemy}}) \cdot \log(N_{\text{obstacle}} + N_{\text{enemy}}))$ to calculate.

**AX0.** This is the same as **A0** except that $N_{\text{coin}} > 1$. This becomes reminiscent of the traveling salesman problem (TSP). To simplify the discussion, let us assume that the agent moves continuously with speed $v_{\text{agent}}$ instead of discretely with step size $v_{\text{agent}}$, and assume that the each coin's value decreases to a factor of $\gamma^t$ at time $t$. Let us label the agent as $0$ and label the coins from $1$ through $N_{\text{coin}}$, with corresponding values $v_1, v_2, \cdots, v_{N_{\text{coin}}}$. Let the Manhattan distance between object $i$ and $j$ be $d_{i,j}$. Let $c_0 = 0$. Then the optimal policy corresponds to determining a permutation of $\{1, 2, \cdots, N_{\text{coin}}\}$, denoted by $(c_1, c_2, \cdots, c_{N_{\text{coin}}})$, to minimize

$$\sum_{i=1}^{N_{\text{coin}}} v_{c_i} * \gamma^{\sum_{k=1}^{i} d_{c_{k-1}, c_k}}. \tag{1}$$

We believe this is an NP-hard problem in terms of $N_{\text{coin}}$ due to its resemblance to TSP. Therefore, it is very likely that the optimal policy can only be calculated in time $O(N_{\text{coin}}!)$.

**AX1.** This is the same as **A1** except that $N_{\text{coin}} > 1$. By similar arguments used for **A1** and **AX0**, it is very likely that the optimal policy can only be calculated in time $O(N_{\text{coin}}! + N_{\text{coin}} \cdot N_{\text{obstacle}} \cdot \log(N_{\text{obstacle}}))$.

**AX2.** This is the same as **A2** except that $N_{\text{coin}} > 1$. By similar arguments used for **A2** and **AX0**, it is very likely that the optimal policy can only be calculated in time $O(N_{\text{coin}}! + N_{\text{coin}} \cdot (N_{\text{obstacle}} + N_{\text{enemy}}) \cdot \log(N_{\text{obstacle}} + N_{\text{enemy}}))$.

**B0, B1, B2.** These corresponds to variants of **A0, A1, A2** where $N_{\text{enemy}} > 0$, $v_{\text{enemy}} > 0$, and the enemies follow the default movement protocol. These variants should be harder and more complex than their original version since now the shortest path keeps changing due to the movements of the enemies. When $N_{\text{enemy}}$ is relatively small, a path finding algorithm with simple heuristics to avoid the enemies should be a good candidate for good policies; however, as $N_{\text{enemy}}$ becomes larger, the task becomes overwhelmingly difficult, and even impossible if $N_{\text{enemy}}$ is too large (because in these cases the agent does not have means to eliminate the enemies).

**BX0, BX1, BX2.** These corresponds to variants of **B0, B1, B2** where $N_{\text{coin}} > 1$. These variants should be harder and more complex than their original version since now the challenge also involves certain aspect of minimizing (1) as discussed in **AX0**, the distance between any two objects could be

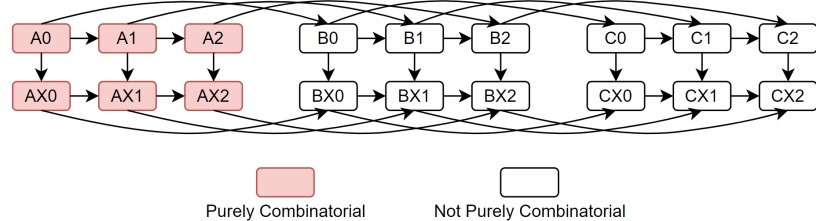

Figure 2: Configurations of different decision complexities. Arrow points from lower decision complexity to higher decision complexity.

ever changing as discussed in **B0, B1, B2**. The same conclusion we had for **B0, B1, B2** also applies here.

**C0, C1, C2.** These corresponds to variants of **B0, B1, B2** where $N_{\text{bomb}} > 0$ and $N_{\text{projectile}} > 0$. These variants should be more complex than their original version in terms of decision making since now the agent can use offensive tools (bombs and projectiles) to eliminate both the enemies and the obstacles. However, in terms of the difficulty of the game (i.e., not dying and achieving high scores), these variants are not necessarily harder. The obstacles now can be destructed to open shorter paths to coins, and the enemies can be eliminated so that the agent is less likely to die due to being cornered by enemies as in **B0, B1, B2**.

**CX0, CX1, CX2.** These corresponds to variants of **C0, C1, C2** where $N_{\text{coin}} > 1$. These variants should be harder and more complex than their original version since now the challenge also involves certain aspect of minimizing (1) as discussed in **AX0**, the distance between any two objects could be ever changing due to both enemy movements and the elimination of enemies and obstacles. The same conclusion we had for **C0, C1, C2** also applies here. We summarize the configurations discussed above in Figure 2. As we have seen so far, the presence of moving enemies, projectiles, and bombs distinguishes Arena from pure combinatorial optimization problems, and opens the possibilities for learning-based approaches to obtain good policies. Of course, there are many other cases; for example, we could set $N_{\text{enemy}}$ to 0 and set $N_{\text{obstacle}}, N_{\text{bomb}}, N_{\text{projectile}}$ to all be greater than 0. This way a good policy needs to be able to "terraform" the map to create short-cuts to coins, but does not need to have any capacity to avoid or eliminate enemies. In the most extreme case, all features of the Arena are enabled and the enemies are equipped with adversarial protocols, which can easily be a huge challenge for any existing policy learning algorithms.

## 5 Experiments

We perform imitation learning experiments under configurations **BX2**, **CX2** and reinforcement learning experiments under configuration **AX0** to demonstrate how policy learning can be performed on the Arena benchmark.

### 5.1 Methodology

We consider two types of policies: heuristic policies and learning based policies. Heuristic policies are essentially hand-designed (algorithmic) rules. These policies can give intuitive understanding about the difficulty of an environment and are important baselines for learning-based policies. Learning-based polices are policies obtained through machine learning, e.g., reinforcement learning and imitation learning. We will use heuristic policies to generate demonstrations for imitation learning and perform reinforcement learning from scratch.

#### 5.1.1 Heuristic Policies

**BX2.** Under this configuration, the agent is not able to place bombs or fire projectiles. Therefore, the agent has to avoid enemies and detour around obstacles. In any state, let $\text{sp}(i, j)$ be the length of the shortest path between object $i$ and $j$ (avoiding current enemy locations and obstacles), let $u$ be any potential coordinate of the agent in the next step if it chooses to move, the decision of the agent relies

on the following heuristic function

$$h(u, \mathcal{E}, \mathcal{C}, \mathcal{O}) = \min_{c \in \mathcal{C}} \mathrm{sp}(u, c) + \sum_{e \in \mathcal{E}} \frac{1}{\mathrm{sp}(u, e)}.$$

Let $u$ be a minimizer of $h$ and $d$ the corresponding direction of the movement. The agent chooses movement $d$ as its action. Intuitively, the agent simply follows the shortest path and avoid obstacles and enemies to collect the closest coin. We also make sure that its behavior is slightly conservative by making it deliberately keep away from the enemies.

**CX2.** This configuration is different from **BX2** considered above, in that the agent now has the choice to eliminate enemies and obstacles using either bombs or projectiles. Let Manhattan$(i, j)$ be the Manhattan distance between object $i$ and $j$, and let $u$ be any potential coordinate of the agent in the next step if it chooses to move. The decision of the agent relies on the following two heuristic functions

$$h_1(u, \mathcal{E}, \mathcal{C}, \mathcal{O}) = \min_{c \in \mathcal{C}} \mathrm{Manhattan}(u, c) + \sum_{e \in \mathcal{E}} \frac{1}{\mathrm{Manhattan}(u, e)},$$

$$h_2(u, \mathcal{E}, \mathcal{O}) = \frac{1}{\{\text{number of steps to hit an enemy or a obstacle if firing a projectile}\}}.$$

Let $u$ be a minimizer of $h_1$ and $d$ the corresponding direction of the movement. If $d$ is the same direction as the direction the agent is facing, it fires a projectile with probability $h_2(u, \mathcal{E}, \mathcal{O})$. If the agent does not fire a projectile, it chooses movement $d$ as its action. Intuitively, the agent always moves towards the closest coin (while slightly avoids enemies) in terms of the Manhattan distance and destroys enemies and obstacles along the way.

### 5.1.2 Learning-based Policy

We perform imitation learning on all the configurations and reinforcement learning on configuration **AX0**. In configurations other than **AX0**, our DQN is not able to learn a policy of reasonable performance within 10 hours. Due to space limitation, for imitation learning, we only present results on configurations **BX2, CX2** here (the rest results are on project webpage). We use the average score from 100 runs to measure the performance of a policy.

**Imitation Learning.** We first describe the training configurations. For both **BX2** and **CX2**, we choose $H = W = 128$, $S = 8$, $N_{\mathrm{coin}} \sim \mathrm{uniform}(\{1, 2, \cdots, 5\})$, $N_{\mathrm{enemy}} \sim \mathrm{uniform}(\{0, 1, \cdots, 5\})$, $N_{\mathrm{obstacle}} \sim \mathrm{uniform}(\{0, 1, \cdots, 10\})$, $v_{\mathrm{agent}} = v_{\mathrm{enemy}} = 2$, $\delta = 0.01$, $\gamma = 0.99$, and the size of all obstacles are set to 16. For **CX2**, we additionally set $N_{\mathrm{projectile}} = N_{\mathrm{bomb}} = 3$, $D_{\mathrm{bomb}} = 100$, $r_{\mathrm{bomb}} = 32$, $v_{\mathrm{projectile}} = 8$. The heuristic policy is used to collect 300,000 states through the interaction with the environment, and label them with the taken actions. To perform imitation learning, we convert the state representation given by Arena into either a *complete graph*, in which each vertex (an object or the agent) is connected to the rest vertices, or a *star graph*, in which only the vertex corresponding to the agent is connected to other vertices (objects). Each vertex is associated with a vector attribute containing the corresponding object's type, coordinate, velocity, and bounding box (the region it occupies). With the graph representation of the states, a graph neural network whose architecture is described in (Wang et al., 2019) is trained on the collected demonstrations. We train the network for 200 epochs with batch size 32 and weighted cross-entropy loss, where the weight is inversely proportional to the frequency each label appears in the demonstration.

**Reinforcement Learning.** We perform reinforcement learning on the configuration **AX0**. Training an RL agent from scratch is considerably harder than performing imitation learning due to (1) the lack of demonstrations generated by the heuristics policy (2) the use of a sparse reward signal. We choose $H = W = 64$, $S = 8$, $N_{\mathrm{coin}} \sim \mathrm{uniform}(\{1, 2, \cdots, 5\})$, $v_{\mathrm{agent}} = v_{\mathrm{enemy}} = 2$, $\delta = 0.01$, $\gamma = 0.99$, and no enemies or obstacles. The agent will receive a $+1$ reward when it reaches a coin. We utilize a DQN agent (Mnih et al., 2013) built on top of a graph neural network similar to the one used in the imitation learning experiment. We use the *complete graph* setup, train the agent for 5,000 episodes with batch size 64 and a learning rate of $1e - 4$. We exponentially decay the exploration rate $\epsilon$ from 0.9 to 0.05 with a rate of 0.995. We use a replay buffer of size $1e5$.

## 5.2 Results

**Imitation Learning on BX2.** As shown in Table 3, the result shows that our GNN model can approximate the heuristic algorithm and possess certain compositional generalizability. However, there is still a gap between our model and the teacher policy.

**Imitation Learning on CX2.** As illustrated in Table 4, GNN using state representation of both *complete graph* and *star graph* can achieve comparable result to heuristic policy on training scenarios ($N_{coin} = 1, 3, 5$). In terms of unseen scenarios ($N_{coin} = 7, 9$), there is a performance gap between heuristic policy and GNN with *complete graph*, but surprisingly GNN with *star graph* performs better than its heuristic teacher.

Table 3: Imitation learning on **BX2**. We use the same demonstration dataset generated by rolling out the heuristic policy on training scenarios ($N_{coin} \sim \text{uniform}(\{1, 2, \cdots, 5\})$, $N_{enemy} \sim \text{uniform}(\{0, 1, \cdots, 5\})$, $N_{obstacle} \sim \text{uniform}(\{0, 1, \cdots, 10\})$) for behavior cloning. All the GNN results are obtained from the same cloned GNN policy.

| $N_{coin}$ | $N_{enemy}$ | $N_{obstacle}$ | Heuristic policy | Star graph (GNN) |
|---|---|---|---|---|
| 1 | 1 | 2 | $\mathbf{0.683 \pm 0.161}$ | $0.669 \pm 0.170$ |
| 3 | 3 | 6 | $\mathbf{1.686 \pm 0.391}$ | $1.420 \pm 0.452$ |
| 5 | 5 | 10 | $\mathbf{2.299 \pm 0.518}$ | $1.873 \pm 0.852$ |
| 7 | 7 | 14 | $\mathbf{2.515 \pm 0.924}$ | $1.917 \pm 1.243$ |
| 9 | 9 | 18 | $\mathbf{2.167 \pm 0.769}$ | $1.683 \pm 1.640$ |

Table 4: Imitation learning on **CX2**. We use the same demonstration dataset generated by rolling out the heuristic policy with the same training setup as Table 3 for behavior cloning. All the GNN results are obtained from the same cloned GNN policy.

| $N_{coin}$ | $N_{enemy}$ | $N_{obstacle}$ | Heuristic policy | Complete graph (GNN) | Star graph (GNN) |
|---|---|---|---|---|---|
| 1 | 1 | 2 | $\mathbf{0.677 \pm 0.161}$ | $0.663 \pm 0.186$ | $0.671 \pm 0.170$ |
| 3 | 3 | 6 | $\mathbf{1.637 \pm 0.559}$ | $1.630 \pm 0.559$ | $1.666 \pm 0.452$ |
| 5 | 5 | 10 | $2.295 \pm 0.853$ | $2.334 \pm 0.838$ | $\mathbf{2.335 \pm 0.852}$ |
| 7 | 7 | 14 | $2.869 \pm 1.213$ | $2.683 \pm 1.327$ | $\mathbf{2.861 \pm 1.243}$ |
| 9 | 9 | 18 | $2.650 \pm 1.608$ | $2.153 \pm 1.640$ | $\mathbf{3.052 \pm 1.546}$ |

**Reinforcement Learning on AX0.** The results are shown in Table 5. The DQN agent is trained with $N_{coin}$ uniformly sampled from 1 to 5. The learned policy is evaluated on environments with $N_{coin}$ in a larger range. We run evaluation for each $N_{coin}$ for 100 times and report the average score as well as the standard deviation.

Table 5: Reinforcement learning (DQN) on **AX0**

| $N_{coin}$ | 1 | 3 | 5 | 7 | 9 | 11 |
|---|---|---|---|---|---|---|
| Score | $0.9 \pm 0.3$ | $2.87 \pm 0.439$ | $4.93 \pm 0.515$ | $6.58 \pm 1.408$ | $4.15 \pm 4.405$ | $2.17 \pm 4.171$ |

## 6 Future Work

As one of the first policy learning benchmarks that focus on scalability and configurability, we have kept the mechanics of the environments to be relatively simple. However, the Arena benchmark can be easily extended to cover a much broader spectrum of game logic, such as causal inference, cargo transportation, resource harvesting. It can also be extended to have a multi-player support for testing collaborative or competitive multi-agent decision making.

While we have performed some preliminary experiments, we expect more experiments, especially reinforcement learning experiments, to be done on the Arena benchmark. It would also be very interesting to explore different model choices such as models that are hybreds of GNNs and algorithms.

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
