# OpenReview forum: "Arena: A Scalable and Configurable Benchmark for Policy Learning"
_NeurIPS.cc/2021/Track/Datasets_and_Benchmarks/Round1 — Submitted to NeurIPS 2021 Datasets and Benchmarks Track (Round 1)_

### Official Review · Reviewer_DNYt · 2021-07-03
**An open source Python game for policy learning.**

**Rating:** 4
**Confidence:** 5

**Strengths:**

* The Arena game complexity was analyzed thoroughly.
* The Arena game is open source.



**Weaknesses:**

* The authors emphasized that Arena is a scalable and configurable benchmark and the state space in Arena is object-based. However, these properties do not seem to be unique to Arena. The StarCraft II benchmark, to the best of my knowledge, is also scalable and configurable and its state space is object-oriented. In many other benchmark environments, if the state space is vector-based, it might be converted to object-based, allowing the policy learning using graph neural networks. Many existing open source Python games, such as [WaterWorld](https://pygame-learning-environment.readthedocs.io/en/latest/user/games/waterworld.html), are configurable and scalable. With some modifications or extension to the source code, it should be easy to obtain the object-based state representation as well.

* Arena is a perfect-information game. However, when the game gets "scaled up" by adding more objects to it, the perfect-information setting is making less sense. For example, suppose there are 10,000 enemies in the game, the agent would not need to know the enemy that are extremely far away, and those enemies should not contribute to the complexity of computing the optimal policy. At least an option of imperfect-information game setting is required.

* In chapter 5.1.2, "In configurations other than AX0, our DQN is not able to learn a policy of reasonable performance within 10 hours." This is not satisfying. According to chapter 3.4, with a 3.60 GHz CPU, Arena could simulate 2,300 time steps per second for 100 objects in the environment, which means the environment simulation should not be a bottleneck for training. As a benchmark, the authors should have the responsibility to provide reasonable performance baseline for each configuration using baseline model. If DQN is not a suitable algorithm given the complexity of game state, the author should also try using other algorithms.


**Additional Feedback:**

* I have complex feelings when I see Arena.
* Many existing games, especially Python games, could be "transformed to" Arena without too much effort. This eliminates the novelty and importance of Arena as a "dataset" or benchmark.
* The baseline analysis should be more thorough.


**Clarity:**

* In chapter 4, when analyzing the complexity of A1 and A2, the author should elaborate on the graph algorithm used for the shortest path problem. Did you assume the H and W are constant values therefore they do not appear in the big-O notations?
* The authors did not mention the baseline experiment details, such as if they have used GPU for training the baseline models. In the meantime, they claim DQN is not able to learn well in 10 hours for configurations other than AX0. The reader will have no idea whether this "10 hours" is with CPU training or GPU training, if it were GPU, how many GPUs were used, how data parallel was used for training DQN.
* The authors should provide some insights about the "surprising" results seen in the Table 4.

**Correctness:**

* In the Table 4, N_coin = 7, the heuristic policy result is better than the graph neural network results. But the author claims that "In terms of unseen scenarios (N coin = 7, 9), there is a performance gap between heuristic policy and GNN with complete graph, but surprisingly GNN with star graph performs better than its heuristic teacher."

**Documentation:**

* The Arena is open source on GitHub.
* I would like to see the scripts to fully reproduce results in Table 3 and Table 4 on GitHub.

**Ethics:**

* No.

**Relation To Prior Work:**

* The authors clearly discussed how this work differs from previous contributions.
* However, the author only mentioned many widely-used and famous environments, which are somewhat difficult to configure and/or be scaled. There are pure open source, although less well known, Python game environments, similar to the Arena that the authors created, that can be configured and scaled. The authors should have done more thorough searches on those, analyze and compare the Arena with those.

**Summary And Contributions:**

* The authors submitted an open source Python game, named Arena, for policy learning. The game is configurable and scalable and the user could change the game complexity by tuning the game parameters. For the user's convenience, especially for graph learning, the game state is represented using graph data structure.
* Some preliminary imitation learning and reinforcement learning experiments were conducted for Arena.
* Decision remains unchanged on July 19th, 2021.

---

> ### Author Response · Authors · 2021-07-15
> **Response to Reviewer DNYt**
>
> Regarding the option of imperfect-information game setting, we want to clarify that, our intention is to give users complete game information, effective information can be chosen by users themselves or by neural networks such as attention modules. We agree with the reviewer that the difficulty of the game does not necessarily increase linearly with the number of objects. This property may help neural networks solve this task. We will conduct experiments with imperfect-information setting in the next submission.
>
> Regarding other suggestions, we thank the reviewer for valuable feedback. We will make the suggested change in the next submission.

---

> > ### Comment · Reviewer_DNYt · 2021-07-20
> > **Thank you.**
> >
> > Thank you very much. I am looking forward to your next submission.

---

### Official Review · Reviewer_eHAi · 2021-07-04

**Rating:** 6
**Confidence:** 4

**Strengths:**

1. The benchmark fills a useful niche of a testbed for graph neural networks (though I am not that familiar with the related work, so I may be missing some other benchmark that already serves this purpose).
2. It seems very easy to scale up the size and difficulty of the environment -- just change a few numeric parameters -- which, in addition to being useful for GNNs, could also be useful for developing a curriculum.

**Weaknesses:**

1. As mentioned below in Correctness, the claims about time complexity seem overstated.
2. The paper is missing some discussion of related work (discussed below).
3. I wish there was more motivation for the specific choices made in the benchmark. Why use Bomberman-style dynamics?

**Additional Feedback:**

N/A

**Clarity:**

Yes, the paper was overall well written and easy to understand. One recommendation would be to move the detailed, mathematical description of the environment dynamics in Section 3.2 to the appendix, and provide a slightly higher-level, English description of the dynamics in the main paper.

**Correctness:**

The paper is not very careful in its use of time complexity, and ends up making incorrect statements as a result. For example, on AX0, the authors say:

> Therefore, it is very likely that the optimal policy can only be calculated in time O(Ncoin!).

This seems too strong. I believe that the problem is probably NP-hard (and a budget version would be NP-complete), but most such problems can be solved in roughly exponential time (which is much faster than factorial time). Indeed, Traveling Salesman can itself be solved in exponential time by doing dynamic programming on all possible subsets of destinations: I think a similar approach could work in this case as well.

(If we want to get really pedantic the claim is likely correct, since $O(n!)$ means $n!$ time _or less_. However, it’s clear that in context the authors really mean $\Omega(n!)$, i.e. $n!$ time _or more_, and that’s what I expect readers to take away.)

I have similar worries about the discussion for later classes. In general I recommend being careful about claiming lower bounds on required time complexity -- it is often quite difficult to establish such bounds. (NP-hardness is a special case where it is often relatively easy to establish, though it can still be tricky.) For example, the graphs in Arena will satisfy the triangle inequality -- are you sure there isn’t an algorithm that can leverage this fact to do better? (There is in fact a special algorithm for Traveling Salesman on graphs that satisfy the triangle inequality, though it is only an approximation algorithm, rather than one that finds the exact solution.)

Nits:

In Table 4, the star graph should be bolded in the second row, and the heuristic policy should be bolded in the fourth row. The text in lines 314-318 should be updated appropriately. Also, given how noisy the results are, claims such as “surprisingly GNN with star graph performs better than its heuristic teacher” should be marked as speculative.

**Documentation:**

Documentation looks good.

**Ethics:**

No ethical concerns.

**Relation To Prior Work:**

There is a good amount of discussion of existing RL benchmarks. However, there are other benchmarks, particularly in multiagent RL, that the authors should be aware of. In particular, the Bomberman-style game dynamics have been used in the [Pommerman competition](https://arxiv.org/abs/1809.07124), which should be discussed in the paper. In addition, the name “Arena” has already been used (recently) in a [multiagent RL benchmark](https://arxiv.org/abs/1907.09467); it should not be reused again so soon and in such a related area.

**Summary And Contributions:**

The paper proposes a benchmark for reinforcement learning with dynamics similar to those in the game of Bomberman. The core feature of this benchmark is that it is _scalable_: it is possible to increase the size of the map, change the number of obstacles, enemies, or coins, etc. This is particularly useful to test graph neural networks (GNNs), which can handle arbitrary numbers of objects represented as a graph. The authors run some experiments with reinforcement learning and imitation learning to demonstrate the use of the benchmark with GNNs.

---

> ### Author Response · Authors · 2021-07-15
> **Response to Reviewer eHAi**
>
> Regarding the time complexity, we want to clarify that, our intention of discussing time complexity is to provide intuition about the difficulty level of environment variants. We do not mean to claim rigorously that the algorithm mentioned is optimal. We agree with the reviewer that better algorithms that exploits property of Arena may exist. We will rephrase those parts in the next submission to better reflect our intentions. Note that AX0(collecting multiple coins) is slightly different from TSP in that nodes can be visited more than once. So what matters is the order of coins, which leads to $O(N_{coin}!)$ time complexity.
>
> Regarding related works, we thank the reviewer for pointing out relevant works that we are not aware of. We will include discussion about them in related works in the next submission.

---

> > ### Comment · Reviewer_eHAi · 2021-07-15
> > **Pretty sure the TSP algorithm still applies**
> >
> > > Note that AX0(collecting multiple coins) is slightly different from TSP in that nodes can be visited more than once.
> >
> > This can be handled by a preprocessing step in which you find the shortest distance from node i to node j (which may go through some other nodes).
> >
> > > So what matters is the order of coins
> >
> > This is also true in TSP. The dynamic programming keeps track of the _set_ of coins visited already, as well as the _last_ coin visited; it can effectively deal with the order as well because of how it keeps track of the last coin.
> >
> > For more details see https://www.baeldung.com/cs/tsp-dynamic-programming

---

### Official Review · Reviewer_gEM2 · 2021-07-05
**Arena: A promising environment in search of a motivating question**

**Rating:** 4
**Confidence:** 4

**Strengths:**

[S1] Flexibility: the environment could be used to evaluate generalized policy learning, curriculum generation strategies, multiagent learning (by controlling the enemies), and probably various things I haven't thought of. "Remixes" of this environment might be quite valuable to the community.

[S2] Discussion of environment mechanics is thorough (although having all of this in the body of the paper impedes clarity—see below).

[S3] Each environment configuration comes with a hand-coded baseline policy and analysis of its complexity. This makes it obvious that the environments are solvable in principle, and suggests what kinds of "skills" an agent needs to have to solve them. This kind of analysis is very useful, but unfortunately lacking in most deep RL environments.

**Weaknesses:**

[W1] The biggest weakness of this paper is the lack of a motivating question for the benchmark. What is the "big question" that this benchmark is meant to answer? e.g. is it the performance of different algorithms for learning generalised policies? If so, what kind of generalisation? (interpolation, combinatorial generalisation, something else?)

Flexibility is great, but it's also important to have a guiding question that (1) convinces other researchers that the benchmark will help them investigate something valuable (or showcase their cool algorithm), and (2) provides guidance to future authors on how to use the benchmark, so that results between papers that use the benchmark are comparable. The lack of a big motivating question made it difficult to determine whether the benchmark is appropriate for its goals, whether it contributes something over existing work, whether the evaluation is appropriate, etc.

[W2] A related weakness is that the experiment section does not lay out any clear hypotheses to be tested. It presents results for $N$ algorithms on $M$ environments, but it's not clear what the reader is meant to take away from this—what did we learn about AI from these experiments? This is discussed further under correctness.

[W3] Various clarity issues, including verbosity in the main body of the paper and a somewhat uninformative related work section. This is discussed further in the clarity section.

**Additional Feedback:**

I think this environment has potential, but the paper needs more clarity around what question the benchmark is intended to answer. Without this, I think it will be hard to convince the community to adopt it. This is the reason for my low overall rating.

-----

Update: Thank you to the authors for their response. I'm going to keep my score as-is—good luck on the future submission!

(review last checked/updated 2021-07-19)

**Clarity:**

The paper is generally quite precise, although I had some concerns about readability:

[CL1] The detailed dynamics in Section 3 are great for reproducibility. However, it was a bit overwhelming to have all these details dropped on me at this point in the paper. It would be more helpful to have a brief, intuitive description of how the various game mechanics function, with the gory details saved for an appendix.

[CL2] Section 4 introduces various categories of environment configuration (A0, BX1, CX0, etc.). It was hard for me to understand the abbreviated codes used for these environments. Is it possible to replace the letters and numbers with mnemonics? e.g. you could replace "A" with "Obstacles only" or "OO", "B" with "Obstacles, enemies" or "OE", and "C" with "Everything" or "Ev". This would save the reader from constantly flipping back to determine what the various codes mean.

[CL3] A few design decisions weren't super well-motivated. e.g. I'm not sure what the default behaviour of enemies is to move randomly—doesn't that seem a bit easy to evade, relative to enemies that seek the player out? It's also not clear to me why the geometrically decreasing value of the coins is baked into the dynamics, when the reward function could be geometrically discounted at the algorithm level instead. Finally, the categories in Section 4 seemed somewhat arbitrary. Why were these categories chosen instead of other ones? e.g. why introduce bombs and projectiles at the same time, instead of just one thing at a time?

**Correctness:**

[CO1] My main concern here is the appropriateness of the experimental evaluation, which I touched on in weakness [W2] above. As-is, it's not clear what question the paper is trying to ask. Without a clear question, it's unclear whether the experiments provide a satisfactory answer. I think there are a lot of interesting questions that _could_ be asked here: for instance, do GNNs generalise better across different sizes of environment than models based on self-attention? Benchmarks are valuable to the extent that they help researchers answer interesting questions, so choosing appropriate questions for the experiments section is essential to convincing the reader that there is value to the benchmark.

**Documentation:**

Documentation on Github seems sufficient. I had a cursory inspection of the code and it was generally readable.

**Ethics:**

No.

**Relation To Prior Work:**

[R1] The related work section name-drops several similar papers without actually discussing how they relate to this work. Ideally the related work section should explain why this benchmark is better than existing ones at answering whatever question it is designed to answer. As far as generalized policies are concerned, I'm particularly interested to know why recent benchmarks like ProcGen or MineRL do not provide enough "variation" relative to this benchmark (I think this was the gist of line 27 in the intro).

**Summary And Contributions:**

This paper proposes an extensively parameterised Bomberman-like environment for decision-making. Its parameters can be used to change the size of the environment, to modify the behaviour of objects in the environment in various ways, and to add or remove entire environment mechanics (e.g. removing a class of objects). In principle, this benchmark could be used to evaluate generalised policy learning, although I suspect it's flexible enough to evaluate various other things as well (e.g. methods for constructing curricula).

---

> ### Author Response · Authors · 2021-07-15
> **Response to Reviewer gEM2**
>
> [W1, W2, CO1] Please refer to the general response.
>
> [CL1, CL2] Thanks for the suggestion. We will make the suggested change in the next submission.
>
> [CL3] We will cover motivation of each setting and provide experimental results for each setting.
>
> [R1] We will expand the related works section and include the discussion as suggested.

---

### Author Response · Authors · 2021-07-15
**Common Response to Reviewers**

We would like to thank the reviewers for their helpful feedback and for their time and effort. We plan to overhaul the writing and experiments and submit to a future venue.

We agree with the reviewers that the motivation should be presented clearly, and experiments should be better-structured and more thorough. Arena is a testbed we design to study compositional generalization of policy learning. We will improve writing and reflect the motivation in our next submission. We will also conduct experiments more systematically.

---

### Decision · Program_Chairs · 2021-07-26

**Decision:**

Reject

**Comment:**

The paper proposes a configurable environment for decision-making. The parameters of the environment that are configurable are the parameters like size of the environment and number of objects.

The reviewers pointed the experiments are not sufficient and they are rushed (not well thought enough.) It seems like the writing of the paper requires more work as well. The motivation is lacking. The authors of the paper also suggested that they will overhaul the writing and experiments to resubmit the paper to another venue.